# Density Functional Method Study on the Cooperativity of Intermolecular H-bonding and π-π^+^ Stacking Interactions in Thymine-[C_n_mim]Br (*n* = 2, 4, 6, 8, 10) Microhydrates

**DOI:** 10.3390/molecules27196242

**Published:** 2022-09-22

**Authors:** Yanni Wang, Chaowu Dai, Wei Huang, Tingting Ni, Jianping Cao, Jiangmei Pang, Huining Wei, Chaojie Wang

**Affiliations:** 1The Third Affiliated Hospital of Wenzhou Medical University, The Pharmacy Department of Ruian People’s Hospital, Wenzhou 325000, China; 2The Pharmacy Department of Ruian Hospital of Traditional Chinese Medicine, Wenzhou 325000, China; 3School of Pharmaceutical Sciences, Wenzhou Medical University, Wenzhou 325035, China

**Keywords:** thymine, ionic liquids, cooperativity, H-bonding, π-π^+^ stacking

## Abstract

The exploration of the ionic liquids’ mechanism of action on nucleobase’s structure and properties is still limited. In this work, the binding model of the 1-alkyl-3-methylimidazolium bromide ([C_n_mim]Br, *n* = 2, 4, 6, 8, 10) ionic liquids to the thymine (T) was studied in a water environment (PCM) and a microhydrated surroundings (PCM + *w*H_2_O). Geometries of the *mono*-, *di*-, *tri*-, and *tetra*-ionic thymine (T-*w*H_2_O-*y*[C_n_mim]^+^-*x*Br^−^, *w* = 5~1 and *x* + *y* = 0~4) complexes were optimized at the M06-2X/6-311++G(2*d*, *p*) level. The IR and UV-Vis spectra, QTAIM, and NBO analysis for the most stable T-4H_2_O-Br^−^-1, T-3H_2_O-[C_n_mim]^+^-Br^−^-1, T-2H_2_O-[C_n_mim]^+^-2Br^−^-1, and T-1H_2_O-2[C_n_mim]^+^-2Br^−^-1 hydrates were presented in great detail. The results show that the order of the arrangement stability of thymine with the cations (T-[C_n_mim]^+^) by PCM is stacking > perpendicular > coplanar, and with the anion (T-Br^−^) is front > top. The stability order for the different microhydrates is following T-5H_2_O-1 < T-4H_2_O-Br^−^-1 < T-3H_2_O-[C_n_mim]^+^-Br^−^-1 < T-2H_2_O-[C_n_mim]^+^-2Br^−^-1 < T-1H_2_O-2[C_n_mim]^+^-2Br^−^-1. A good linear relationship between binding *E*_B_ values and the increasing number (*x* + *y*) of ions has been found, which indicates that the cooperativity of interactions for the H-bonding and π-π^+^ stacking is varying incrementally in the growing ionic clusters. The stacking model between thymine and [C_n_mim]^+^ cations is accompanied by weaker hydrogen bonds which are always much less favorable than those in T-*x*Br^−^ complexes; the same trend holds when the clusters in size grow and the length of alkyl chains in the imidazolium cations increase. QTAIM and NBO analytical methods support the existence of mutually reinforcing hydrogen bonds and π-π cooperativity in the systems.

## 1. Introduction

Ionic liquids (ILs) have been widely used in many fields due to their large variety and unique properties, which are a class of liquid compounds consisting of a bulky organic cation and relatively small anion and provided with excellent properties such as very low vapor pressure, nonvolatility, nonflammability, high thermal and electrochemical stability, and have been touted as greener alternatives to the conventional organic solvents [1,2,3]. However, the ILs create hazards to the natural environment and various biological levels of organization [4,5,6], which are indicated by the inhibition of enzymatic activity (e.g., acetylcholinesterase [7,8,9], the antioxidant enzyme of the animal liver [10,11,12]), the antimicrobial activity (e.g., *Vibrio fischeri* [13], *Escherichia coli* [14,15]), the phytotoxicity to aqueous and terrestrial plants [16,17] (e.g., *Lemna minor* [18], *Ulva lactuca* [19], *Hordeum vulgare* [20,21]), and the toxicity to invertebrates (e.g., *Daphnia magna* [13,22,23], *Caenorhabditis elegans* [24]) as well as vertebrates (e.g., *Rana nigromaculata* [25], *Danio rerio* [26], rat pheochromocytoma PC12 [27,28], even to human cell lines including human colon carcinoma CaCo-2 and HT-29 [29], IPC-81 [30], human cervical cancer HeLa [31], human lung carcinoma A549 [32], and so on. Furthermore, it was found that all the tested systems have shown distinctive susceptibilities by ILs at different levels. Among all ILs, the 1-alkyl-3-methylimidazolium salts are one of the most pervasive-studied ILs [33,34] reported toxicologically, which showed oxidative stress and genotoxicity in separate living beings and are also a broad subject as bioavailable pollutants [4,8,22,33,35,36]. DNA macromolecule, the genetic information carrier for living organisms and numerous viruses, is witnessing the damage caused by the imidazolium ILs, which occurs concomitantly to a cascade of biological consequences appearing in the cells, organs, and even the whole population [37,38,39]. Nonetheless, the molecular mechanisms of ILs-initiated potential impacts on organisms are still rarely understood.

The theoretical calculation can explain the complicated experiments and provides more accurate information about the nature of noncovalent interactions (NCIs) between ILs and DNA, which contributes to understanding the binding characteristics of imidazolium ILs and DNA compared with the binding Gibbs energies of the 1-alkyl-3-methylimidazolium bromide ILs ([C_n_mim]Br, *n* = 4, 6, 8, 10, 12) and DNA using the pyrene probing fluorescence. Wang et al. [40] have considered that the electrostatic interaction between the imidazolium cations and DNA dominates the system, for which the hydrocarbon chain lengths of ILs binding to DNA are also non-negligible. Subsequently, molecular dynamics (MD) simulation data have further designated that [C_n_mim]Br ILs (*n* = 2, 4, 6) occupy the DNA strands as expected [41]. Once indicated earlier [42,43,44], ILs could bind to the minor groove in the form of a water-assisted “IL spine” instead of the classical “hydrated one” termed as an appealing feature of rigidifying the native B-DNA conformation, which tends to take precedence over orientating themselves about ApT-rich sequences in the minor groove region of the double helix [44]. At this juncture, the structures of π-stacking base pairs and the helix DNA are altered to a certain extent [45], which gives reason to state that other interactions could be allowed to appear between the nucleic bases and ILs. Studies on the structures and properties of the nucleobases, hydrated systems, and imidazolium bromide ionic liquids are still increasing [46,47,48,49,50,51]. Many research reports have been published on the structure and properties of the intermolecular interaction system between T and water [52,53,54,55,56,57]. In work [52], the NCI analysis between C-H and H_2_O is meticulous and in-depth. Experimental and theoretical examination of the specific interaction modes between the nucleic acid bases sand ILs are still relatively limited. The present work is intended to explore and elucidate in detail that the interaction modes between 1-alkyl-3-methylimidazolium bromide ILs ([C_n_mim]Br, *n* = 2, 4, 6, 8, 10) and canonical thymine in the diketo form [58], which is the most stable isomer identified in the gaseous or aqueous phase.

## 2. Materials and Methods

It has been found that the hybrid meta-GGA functionals M06-2X [59], including double Hartree-Fock exchange, is one of the best functionals to yield a more practical effect for the description of the noncovalent interactions, such as the π-stacking interactions on the aromatic bases [60,61], hydrogen-bonding (HB) patterns on the single DNA strand [61], and neutral water clusters [62]. This method also reproduces a better prediction of the geometrical properties for the nucleic acid bases. As a matter of fact, in our previous studies [63,64], M06-2X conjugated with 6-311++G(2*d*, *p*) basis has proved to show superior performance for the depiction of intermolecular interactions in 2[C_2_mim]^+^-2Br^−^ among a series of DFT methods as M06L, M06-2X, B3LYP, CAM-B3LYP, LC-ɷPBE, and ωB97XD methods.

In this work, we have first computed and found the most preferred binding modes of thymine with [C_n_mim]Br (*n* = 2, 4, 6, 8, 10) at the M06-2X/6-311++G(2*d*, *p*)/PCM/water level. Based on the most stable configurations existing between thymine and the increased ion clusters, the microhydrated *mono*-, *di*-, *tri*-, and *tetra*-ionic thymine are calculated to explore the cooperativity of the NCIs established between thymine-cation, thymine-anion, and thymine-cation-anion and the effects of the intermolecular NCIs on the structures and properties of diketone thymine in aqueous solution.

The optimized configurations recognized as local minima were confirmed by the absence of symmetry constraints and imaginary vibrational frequencies evaluated. To simplify the identification, the involving microhydrated ionic complexes are labeled as T-*w*H_2_O-*y*[C_n_mim]^+^-*x*Br^−^-*k*, where the number *w* of a water molecule is 5 to 1. In contrast, the ion number’s sum (*x* + *y*) varies from 0 to 4. The item *k* denotes the stable configuration for the given *w* and (*x* + *y*). Relative energies (*E*_R_), as showed in Equation (1), zero-point vibrational energy (ZPVE) corrected, is defined as the difference between the energy of the most stable structure T-*w*H_2_O-*y*[C_n_mim]^+^-*x*Br^−^-1 and other forms T-*w*H_2_O-*y*[C_n_mim]^+^-*x*Br^−^-*k* (*k* > 1).
(1)$$E_{R}=E{\left(complex\right)}_{k>1}-E{\left(complex\right)}_{k=1}$$

The binding energy (*E*_B_), Equation (2), is computed as the discrepancies between the total energy of the optimized complexes and the sum of the energies of the isolated fragments that constitute the studied clusters and is considered the basis set superposition error (BSSE) [65] estimated in a manner of the counterpoise (CP) correction [66]. The more negative the *E*_B_ value, the more favorable energetical complexes are.
(2)$$E_{B}=E\left(complex\right)-\sum_{isolated}\left(fragment\right)$$

The geometry of each monomer should be changed once the complexes are formed, leading to deformation energy; namely, *E*_D_ with a positive sign undergoes a structural transformation of the fragment in free and combined state, which contains the $E_{D}^{T}$, the energy difference between thymine frozen in the optimized complex and the corresponding isolated base, defined as Equation (3).
(3)$$E_{D}^{T}=E{\left(T\right)}_{complex}-E{\left(T\right)}_{monomer}$$

Besides optimization and frequency computations, time-dependent density functional theory (TD-DFT) [67], the quantum theory of atoms in molecules (QTAIM) [68], and natural bond orbital (NBO) [69] analysis at the same level are carried out for the further exploration of the nature of these NCIs in T-*w*H_2_O-*y*[C_n_mim]^+^-*x*Br^−^*k* systems in aqueous solution. All the quantum chemistry calculations have been performed using the Gaussian-16 suites of programs [70], assisted by the Multiwfn package [71].

## 3. Results and Discussion

### 3.1. Geometries and Energetics

#### 3.1.1. Isolated Thymine and Cations Varying Alkyl Chain Lengths

The molecular electrostatic potentials (MEPs) for isolated thymine and separated cation ions in [C_n_mim]Br (*n* = 2, 4, 6, 8, 10) are constructed and shown in Figure 1, where the favorable sites of N-H and C=O for thymine illustrate the more positive red regions and negative blue ones, respectively; the Br^−^ anion is expected to have stronger proton affinity and prefers to appear around the N1-H (54.05 kcal/mol) and N3-H (38.30 kcal/mol) of thymine. In contrast, the [C_n_mim]^+^ (*n* = 2, 4, 6, 8, 10) cations are partial to approaching the C2=O7 (−34.62 kcal/mol) and C4=O8 (−35.75 kcal/mol) sites of thymine. In turn, regarding the cations, the positive charges are more evenly dispersed around the hydrogen atoms of head groups (^+^C(2/4/5)-H) and decrease significantly with the increasing length of the alkyl chain, resulting in possible interactions with the Br^−^ anion and the H-bonding acceptors (HBAs) in thymine; we have also considered that the forming HBs of thymine and 1~5 water molecule(s) explicitly acted as both an H-bonding donor and an H-bonding acceptor (shown in Electronic Supplementary Information (ESI) Appendix A). Subsequently, a further understanding of the interactions was obtained by incorporating the growing ionic cluster one-by-one instead of the case-by-case water molecule with the thymine nucleobase. In other words, the most stable T-5H_2_O-1 hydrate will be replaced orderly by the T-4H_2_O-Br^−^, T-3H_2_O-[C_n_mim]^+^-Br^−^, T-2H_2_O-[C_n_mim]^+^-2Br^−^, and T-1H_2_O-2[C_n_mim]^+^-Br^−^ species (vide infra). The van der Waals distance of Br···H (3.05 Å) [72] and O···H (2.72 Å) [73] is an intimation of the H-bonds formation, which tends toward 180° and priors to be above 110° as stated in the IUPAC recommendations 2011 [74]. 

The geometrical parameters and values of the optimized thymine monomer are collected in Appendix A. The parent thymine molecule belongs to the *C_s_* point group symmetry. It provides a planar pyrimidine ring as observed by X-ray [75], followed closely by the optimized results of thymine in the present work which are very close to those calculated early at the HF/4-31G** and B3LYP/6-31G** levels [75]. Even for the bond distances, we note that a mean absolute deviation (MAD) is only 0.014 Å compared with the experimental data. 

#### 3.1.2. Mono-Ionic Thymine

***Mono-anionic T-Br^−^***. Four different configurations of this complex were optimized, shown in Appendix A in decreasing stability order from T-Br^−^-1 to T-Br^−^-4, where the halogen anion placed itself in the front (T-Br^−^-1 and -2) and top (T-Br^−^-3 and -4) sites of the pyrimidine ring. The latter is considerably less stable than the former, with a single HB between the Br^−^ anion and the hydrogen atom of N1-H or N3-H groups in thymine, which indicates that the N-H···Br^−^ HB contact is favored than the π···Br^−^ interaction. Further, the interaction of the Br^−^ anion with the N1-H group is more favorable than the N3-H group as predicted in the MEP analysis. 

***Mono-cationic T-[C_n_mim]^+^***. In terms of the relative position of the [C_n_mim]^+^ cations binding to thymine, the T-[C_n_mim]^+^ systems can be partitioned into three arrangements (Appendix A): stacking (S)—the imidazolium π ring and thymine π ring are nearly parallel, perpendicular (P)—the π rings are almost perpendicular to each other, and coplanar HB (H)—the π rings are essentially coplanar. To each arrangement for each cation, we successfully calculated 11 structures, which are S4, P4 and H3. They can be classified in two configurations according to the interaction sites of the cations on thymine: ^+^C-H···O7 (S1, P1, H1) and ^+^C-H···O8 (S2, P2, H2). The strongest interaction was established by the planes of the imidazolium and pyrimidine ring approximately paralleling each other and accompanying the ^+^C-H···O7 hydrogen-bonding on one end of the complexes, which is quasi-exclusively with the cations increasing alkyl chains. In these T-[C_n_mim]^+^-1 systems, the binding energies follow the trend T-[C_2_mim]^+^-1 > T-[C_4_mim]^+^-1 < T-[C_6_mim]^+^-1 > T-[C_8_mim]^+^-1 > T-[C_10_mim]^+^-1 (*E*_B_ in absolute values: 16.49 > 15.35 < 16.86 > 16.66 > 16.28 kJ/mol), which indicates that the interaction formed between the [C_6_mim]^+^ and thymine is slightly more favorable than the binding of other cations to base.

As for the most stable structures, T-[C_n_mim]^+^-1 and T-Br^−^-1, the interaction of thymine with [C_n_mim]^+^ cations presents weaker base-ion strength than that with Br^−^ anion (cf. Appendix A), which is given an interpretation of a smaller $E_{D}^{T}$ value in the former than the latter, and this is similar to the solvation process of uracil in the [C_2_mim]Ac and [C_4_mim]Ac ILs, respectively [76]. This result is ascribed to a more substantial HB donor (HBD) site of the referred nucleobase than the imidazolium cations [77]. The mono-anionic T-Br^−^-1 followed its complex, chosen for further analysis and discussion of the microhydrated mono-ionic thymine (Appendix A). 

#### 3.1.3. Di-Ionic T-[C_n_mim]^+^-Br^−^

To better understand how thymine interacts with the cations and anion simultaneously, 12 kinds of structures of T-[C_n_mim]^+^-Br^−^ for the given n are considered and shown in Appendix A, including S4, P4, and H4 systems, a total of 60. The thymine and ion pair interactions are divided into stacking, perpendicular, and coplanar arrangements. As expected, the relative stabilities for separating different configurations of T-[C_n_mim]^+^-Br^−^ are not influenced by the BSSE correction, which leads to a slight decrease in the energy differences and the absolute *E*_B_ values. In agreement, the presence of one counter anion favors the stacking isomers formed by the base and the imidazolium cations. In Figure 2, the lowest energy configurations for the stacking isomers (T-[C_n_mim]^+^-Br^−^-1) are accompanied by the ^+^C-H···O HB and include Br^−^ anion side sandwiched between the two aromatic rings, interacting via HBs with N1-H and ^+^C(2)-H, wherein the lengths and angles of HBs range within 2.445~2.450 Å and 131.04~131.45° (^+^C(6)-H···O7), 2.426~2.434 Å and 145.77~146.14° (N1-H···Br^−^), 2.827~2.835 Å, 118.01~118.44° (^+^C(2)-H···Br^−^). On the other hand, the C2=O7 and N1-H bonds in thymine are elongated by 0.003 Å and 0.012 Å, while the ^+^C(6)-H and ^+^C(2)-H bonds are instead shortened by 0.010 Å and 0.000 Å, respectively. It signifies that the thymine base exhibits a higher sensitivity to the H-bonding environment provided by the Br^−^ anion rather than the [C_n_mim]^+^ cations, while the π-π^+^ interactions still dominate in the base-cation contacts. Upon close inspection of the binding energy of thymine with each cation increasing n, the T-[C_6_mim]^+^-Br^−^-1 complex is presented again as a better intermolecular action in this arrangement (*E*_B_ in absolute value follows in the order 54.41 > 53.16 < 54.69 > 54.61 > 53.40 kJ/mol). As expected, the strength of thymine-ions binding for the most stable T-[C_n_mim]^+^-Br^−^-1 complex is strong enough to break the intermolecular interaction for the lowest energy T-2H_2_O-1 microhydrate.

In common with the T-[C_n_mim]^+^ system, as shown in Appendix A, the second most stable configuration is a perpendicular structure (T-[C_n_mim]^+^-Br^−^-5), where the Br^−^ anion allows itself on top of the imidazolium cations and forms N1-H···Br^−^ HB with thymine, and the [C_n_mim]^+^ cations form double ^+^C(2)-H···O7 and ^+^C(6)-H···O7 HBs with thymine in an edge-to-face manner—adding an anion and leading to a greater stabilization T-[C_n_mim]^+^-Br^−^-1, which are differing from T-[C_n_mim]^+^-1 complexes in the position of the HB binding sites between the base and cations, occupying the “sugar edge” site [78] (i.e., N1-H and C2=O7 sites) of thymine. It is also the case for the less stable and coplanar geometries (T-[C_n_mim]^+^-Br^−^-9), which are reminiscent of the coplanar HBs in the A-T base pair involving the “Watson-Crick” site (i.e., N3-H and C4=O8 sites) [79]. As we know from an MD simulation [42], the stabilities of A-T and G-C base pairs in the hydrated ILs are reversed under the premise that the ions do not interact with the HB-forming atoms of A-T pairwise. Compared with the absolute *E*_B_ values, we have found that the superiority of T-[C_6_mim]^+^-Br^−^-5 in the perpendicular complexes (50.46 > 49.37 < 50.54 > 49.90 > 48.95 kJ/mol), while the slight distinctiveness of T-[C_8_mim]^+^-Br^−^-9 in the coplanar series (46.33 > 47.55 < 48.54 < 48.96 > 47.75 kJ/mol). From the *E*_B_ vales, to T-[C_n_mim]^+^-Br^−^-1, 5, 9 complexes, the stable order is stacking > perpendicular > co-planar.3.1.4. Tri-ionic T-[C_n_mim]^+^-2Br^−^.

Given that the thymine nucleobase is more sensitive to the Br^−^ anion than the [C_n_mim]^+^ cations, it is mandatory for the extension to obtain the 1:3 forms T-[C_n_mim]^+^-2Br^−^ complexes, the most stable T-[C_n_mim]^+^-2Br^−^-1 complexes for the independent arrangement are listed in Appendix A. In this scenario, the predominance of the parallel stacking isomers persisted. Compared to the *di*-ionic thymine systems, a little difference is the T-[C_8_mim]^+^-2Br^−^-1, which is the most favorable energetic complex with the altering carbon atom number n (*E*_B_ in absolute value follows in the order 79.08 > 74.42 < 75.04 < 79.67 > 74.29 kJ/mol). But consistently, this trend also holds for the microhydrated T-2H_2_O-[C_8_mim]^+^-2Br^−^-1 in the following Section 3.1.4.

#### 3.1.4. Tetra-Ionic T-2[C_n_mim]^+^-2Br^−^

Shortly, a new dimension in studied complexes is being investigated. To further understand the strength and properties of interactions between thymine and ILs, it is natural to continue to expand the system of the action to 1:2 base-IL forms, T-2[C_n_mim]^+^-2Br^−^ complexes. The most exciting result is that the preponderance of the stacking isomers was an absolute advantage in the three arrangements mentioned above, that is the stable sequence is stacking (S) > Perpendicular (P) > coplanar HB (H), which is shown to be entirely independent of the cationic alkyl chain length and to have the same tendencies as their magnitudes of *E*_BSSE_ values order. In the most stable T-2[C_n_mim]^+^-2Br^−^-1 structures, the thymine nucleobase, sandwiched between the two cationic rings piling up a π^+^···π···π^+^ stacking interaction along with two weaker ^+^C(6)-H···O7 HBs, interacts via a single HB with each Br^−^ anion, which also forms HBs contacts with the cations situated above or below the pyrimidine ring. Upon the intercalation of a second ionic pairwise, the base-anion, base-cation, and cation-anion interactions are enhanced obviously by the additional N3-H···Br^−^, π···π^+^ stacking along with ^+^C(6)-H···O7 and ^+^C(2)-H···Br^−^ HBs, analogous to the order for interaction strengths of those in the T-[C_n_mim]^+^-Br^−^-1 structures; the result of *E*_B_ in absolute values again observed preferential T-2[C_6_mim]^+^-2Br^−^-1 form in the cations increasing n (109.47 > 107.80 < 109.91 > 109.71 > 107.61 kJ/mol).

Summing up, with increasing ionic clusters size, *E*_B_ values show a preferred binding mode between thymine and Br^−^ anion: front > top; and between the nucleobase and each imidazolium cations varying alkyl chain lengths: Stacking (S) > Perpendicular (P) > Coplanar HBs (H). To the most stable T-*y*[C_n_mim]^+^-*x*Br^−^-1 complexes, the thymine nucleobase is geometrically and energetically more sensitive to the H-bonding environment provided by the Br^−^ anion rather than the π-π^+^ stacking imparted by the imidazolium cations, with a weaker ^+^C-H···O7 HB (Figure 3). 

#### 3.1.5. Ionic T-wH_2_O-y[C_n_mim]^+^-xBr^−^-k Microhydrates

In order to further explore the effects of water molecules and the cation and anion of ionic liquid on the geometric structure, electronic structure, spectroscopy and energetics of the thymine, we constructed T-w_i_H_2_O-y[C_n_mim]^+^-xBr^−^ (i = 0, 1, 2, 3, 4, w = 5, 4, 3, 2, 1, y = 0, 1, 2 and x = 0, 1, 2) hydrates.

***Free-ionic T*****-*w_0_H_2_O hydrates (w_0_ = 1, 2, 3, 4, 5)***. The solvation-first-shell effect, including 1–5 explicit water molecules, was approximately considered, and the corresponding structures are presented in Appendix A. The stability of relevant microhydrated complexes decreases from left to right and agrees reasonably with other authors [53,54,55,56,57,75]. The studies on microhydrated thymine have been reported in many works and will thus not be discussed in detail here.

***Mono-ionic T*****-*w_1_H_2_O-Br^−^ hydrates (w_1_ = 1, 2, 3, 4)***. In comparison, the T-*w_0_*H_2_O-1 (*w_0_* = 1, 2, 3, 4, 5) hydrates are always less stable than those of the T-*w_1_*H_2_O-Br^−^-1 counterparts, which can be interpreted as a matter that the charged Br^−^ anion replaces the neutral water molecule and interacts with thymine, resulting in a more favorable system and a better correlation between the calculated *E*_B_ and increasing water molecule numbers *w* (correlation coefficient *R*^2^: 0.919 > 0.894). Satisfactorily, the arrangement of four water molecules is fairly consistent with the NMR experimental results [80] that the thymine and two water molecules form two Ow-H···O7 HBs. At the same time, one Ow-H···O8 HB is provided by the base, and one water molecule leaves a C5=C6 bond. As for the most stable T-4H_2_O-Br^−^-1 complex, four explicit water molecules and one anion hold the first solvation shell, where the distance between the N1-H group and Br^−^ anion is very close to that in a water environment with PCM (2.352 Å for T-4H_2_O-Br^−^-1 and 2.349 Å for T-Br^−^-1), and the N1-H bond is shortened by 0.004 Å (in Appendix A) as a result of the forming N1-H···Br^−^ in T-4H_2_O-Br^−^-1, substituting for the N1-H···Ow HB in T-5H_2_O-1 hydrate.

***Di-ionic T*****-*w_2_H_2_O-[C_n_mim]^+^-Br^−^ hydrates (w_2_ = 1, 2, 3)***. Similarly, the stepwise hydration of the most stable T-[C_n_mim]^+^-Br^−^-1 should be focused on. A good linear relationship was revealed between the *E*_B_ values and increasing water number *w_2_*. The regression coefficients are up to 0.99 with the progressive hydration, irrespective of the cationic alkyl chain lengths (Figure 4a). From this, the progressive microhydration stabilized the ionic T-[C_n_mim]^+^-Br^−^-1 complexes with the increasing number of explicit water molecules; the lipid chain lengths of the [C_n_mim]^+^ cations have an insignificant influence on the stability of the targeted complex, and this is now the case for the similar and small $E_{D}^{T}$ values, which is consistent with the MD finding that the alkyl chain length of the imidazolium cation plays a smaller impact on the DNA stability [41]. In the microhydrated T-3H_2_O-[C_n_mim]^+^-Br^−^-1 complex, compared with the T-4H_2_O-Br^−^-1, one water molecule positioned around the C2=O7 group (one of the “wobble” sites [81]) is replaced by one cation, in parallel stacked configuration. Relative to the isolated base in an aqueous solution, there again, the N1-H strengthens the N1-H, and C2=O7···Br^−^ and ^+^(C-H)···O7 HBs exchanged by the N1-H···Ow and Ow-H···O7 HBs. In Figure 3 and Appendix A, the interaction between thymine and Br^−^ anion is reflected in angles, 142.08~144.11°, and distances, 2.454~2.482 Å of N1-H···Br^−^ in different complexes, which is weaker than that for the T-4H_2_O-Br^−^-1 (2.363 Å and 161.58°). It is also the case for N3-H···Br^−^ HB. These results indicate that a larger separation between thymine and Br^−^ anion is observed as a consequence of the cations participating, embodied in π···π^+^ centroids distances 3.306~3.327 Å. At the same time, Br^−^ interaction is reflected in HBs, the angles in 155.31~160.64° and distances in 2.840~2.882 Å for N1-H···Br^−^, and the variation of N1-H bond length is about 0.011 Å.

Analysis on the absolute *E*_B_ values for the T-3H_2_O-[C_n_mim]^+^-Br^−^-1 complexes given a varying n reveals a similar tendency to the T-[C_n_mim]^+^-Br^−^-1, that the binding intensity can be arranged orderly as follows: T-3H_2_O-[C_2_mim]^+^Br^−^-1 > T-3H_2_O-[C_4_mim]^+^Br^−^-1 < T-3H_2_O-[C_6_mim]^+^Br^−^-1 > T-3H_2_O-[C_8_mim]^+^Br^−^-1 > T-3H_2_O-[C_10_mim]^+^Br^−^-1 (126.09 > 125.90 < 127.37 > 126.52 > 126.32 kJ/mol), among which the T-3H_2_O-[C_6_mim]^+^Br^−^-1 has been discussed by the more favorable arrangement, although to a minor extent.

***Tri-ionic T*****-*w_3_H_2_O-[C_n_mim]^+^-2Br^−^ hydrates (w_3_ = 1, 2)***. Sequentially, the stacking T-[C_n_mim]^+^-2Br^−^-1 complexes have participated in stepwise adding of 1~2 explicit water molecule(s) to fill in the “sugar edge” site and “Watson-Crick” site with the two anions, respectively. The hydrous T-2H_2_O-[C_n_mim]^+^-2Br^−^-1 is worthy of being mentioned and reproduces the same binding energy order as the anhydrous T-[C_n_mim]^+^-2Br^−^-1, which the T-2H_2_O-[C_8_mim]^+^-2Br^−^-1 slightly favored.

***Tetra-ionic T-w_4_H_2_O-2[C_n_mim]^+^-2Br^−^ hydrates (w_4_ = 1)***. Finally, one additional water molecule to the “Watson-Crick” site based on the T-2[C_n_mim]^+^-2Br^−^-1 reproduces the solvent effect described effectively as a hybrid model and simulated the first explicit solvation shell [76]. The T-1H_2_O-2[C_n_mim]^+^-2Br^−^-1 illustrates a perturbation of N3-H···Ow and Ow-H···O7 HBs in T-3H_2_O-[C_n_mim]^+^-Br^−^-1 through a replacement of N3-H···Br^−^ and the second π-π^+^ stacking along with ^+^(C6-H)···O7 HB, which causes much greater stability. Very consistently, there are no noticeable quantitative discrepancies, with the increasing alkyl chain length in cations, the stabilities for the T-1H_2_O-2[C_n_mim]^+^-2Br^−^-1, hydrates to the T-3H_2_O-[C_n_mim]^+^-Br^−^-1 complexes still follow the order: T-1H_2_O-2[C_2_mim]^+^-2Br^−^-1 > T-1H_2_O-2[C_4_mim]^+^-2Br^−^-1 < T-1H_2_O-2[C_6_mim]^+^-2Br^−^-1 > T-1H_2_O-2[C_8_mim]^+^-2Br^−^-1 > T-1H_2_O-2[C_10_mim]^+^-2Br^−^-1, among which the T-1H_2_O-2[C_6_mim]^+^-2Br^−^-1 is the greatest and the most favorable one (137.46 > 134.71 < 137.63 > 136.71 > 136.29 kJ/mol).

As discussed above and in data listed in Appendix A, the order of stability for these ionic thymine hydrates (including the neat hydrated) is as follows: T-5H_2_O-1 < T-4H_2_O-Br^−^-1 < T-3H_2_O-[C_n_mim]^+^-Br^−^-1 < T-2H_2_O-[C_n_mim]^+^-2Br^−^-1 < T-1H_2_O-2[C_n_mim]^+^-2Br^−^-1. The gradual decreasing $E_{D}^{T}$ values are also responsible for the increased stability of thymine in the targeted complexes. These results are in agreement with the MD finding [41,43] that the decreasing fluctuation of DNA bases could be examined as mounting imidazolium ILs contents. The higher the ion numbers (i.e., the sum of *x* and *y*), the more stable the T-*w*H_2_O-*y*[C_n_mim]^+^-*x*Br^−^ microhydrates. Figure 4b shows a well-established link between the *E*_B_ values. Increasing ion number(s) could be observed and is independent of alkyl chain length, indicating a cooperative interaction between thymine and the imidazolium IL in aqueous solutions. It has been pointed out [41,43,82,83] that there are always small amounts of water retained in the first solvation shell of DNA in the hydrated ILs solutions regardless of its concentration. The different configurations found for each ionic hydrate were based on the anhydrous and lowest energy T-*y*[C_n_mim]^+^-*x*Br^−^ complexes. Herein, the most energetically stable one of each system (*viz*. T-4H_2_O-Br^−^-1, T-3H_2_O-[C_n_mim]^+^-Br^−^-1, T-2H_2_O-[C_n_mim]^+^-2Br^−^-1, and T-1H_2_O-2[C_n_mim]^+^-2Br^−^-1) is employed for analysis of the spectral features and electronic structures to explore the thorough nature of intermolecular thymine-ILs interactions.

### 3.2. IR Vibrational Spectra

#### 3.2.1. Isolated Thymine, Cations, and Water

The infrared vibration spectrum of pure thymine, [C_n_mim]^+^ (*n* = 2, 4, 6, 8, 10), and water molecule are shown in Figure 5(a1). There are overlaps for three monomers, apparently, between the higher frequency region, 3000–3300 cm^−1^, and the fingerprint region, 500–1500 cm^−1^, so they are not well suited to follow the thymine-ion(s) interactions. Appendix A has recorded the calculated wavenumbers, intensities of vibrational modes, and the corresponding main contributions to the maxima absorption for the isolated thymine, which are at 3636 and 3600 cm^−1^; the main contribution for these two peaks involves ν(N1-H) (v1 = 3636 cm^−1^, 96%) and ν(N3-H) (v2 = 3599 cm^−1^, 89%), successively. These values of both stretching modes are well coordinated with Singh’s theoretical results (3639 and 3596 cm^−1^) [84] but higher than his experimental values, which may be due to the anharmonicity and electron correlation omitted from our calculation [84,85,86]. In the 1700–1800 cm^−1^ region, the two absorption peaks peculiar to thymine are the most intense, in Figure 5(a2). The spectral congestion arises owing to the controversial assignments of vibrational modes [87] and the occurrence of Fermi resonance [86,88,89]. The contribution of IR absorption observed at 1774 cm^−1^ mainly comes from mode v7 (1774 cm^−1^, 90%), while at 1723 cm^−1^ originates from mode v9 (1723 cm^−1^, 84%). The three v7, v8, and v9 modes dominate in stretching vibrations of ν(C=O) and ν(C=C), and the latter has a lower wavenumber because of the symmetry for the double C=C bond. Additionally, the ν(C2=O7) mode is infallibly in a higher magnitude than the ν(C4=O8) mode perturbed from the conjugation effect. The main component of three vibrational modes located at 1774, 1735, and 1723 cm^−1^, as shown in Appendix A, is attributed mainly to the ν(C2=O7), ν(C4=O8), and ν(C5=C6) vibration in the order. This localization is identical to the assignments proposed by Miles [90] in that the ν(C2=O7) vibrations are placed at 1692 cm^−1^, and the ν(C4=O8) mixed with ν(C5=C6) located at 1657 cm^−1^. Therefore, the modes of thymine vibrations between the two regions, 3300–3700 cm^−1^ and 1700–1800 cm^−1^, would be paid more attention to monitor the structural variance of thymine moiety of the T-*w*H_2_O-*y*[C_n_mim]^+^-*x*Br^−^-1 when interacting with the ILs in aqueous solution, as discussed above.

It needs to be noted that the assignments of the calculated vibrations for the isolated thymine (T) (mode v1, v2, v7, v8, v9, seen in Appendix A) and the studied complexes agree only qualitatively, as mentioned by Thicoipe et al. [54], that there are minimal displacements for each mode upon the presence of explicit water molecules or ion(s).

#### 3.2.2. Ionic T-wH_2_O-y[C_n_mim]^+^-xBr^−^-1 Microhydrates

The graphical spectra for the T-4H_2_O-Br^−^-1, T-3H_2_O-[C_n_mim]^+^-Br^−^-1, T-2H_2_O-[C_n_mim]^+^-2Br^−^-1, and T-1H_2_O-2[C_n_mim]^+^-2Br^−^-1 microhydrates are depicted in Figure 5(a3–a6). Additionally, in Appendix A, the contributions of the primary vibrations of thymine in the hydrated species to the relevant maximum absorption peaks close to the isolated base are recorded.

As shown in Figure 5(a3), the absorption bands of thymine in the T-4H_2_O-Br^−^-1 complex at 3336, 3125 are red-shifted to the corresponding peaks in the isolated base. The main contributor to these bands is primarily assigned to v9 (100%) and v13 (99%), in which the main components are ν(N1-H) and ν(N3-H), respectively. In the case of T-3H_2_O-[C_n_mim]^+^-Br^−^-1 hydrates (shown in Figure 5(a4)), the specific value for each absorption listed in Appendix A embodies an insignificant effect of the imidazolium cations with different n (i.e., the varying length of the alkyl chains) on vibrational wavenumber for the *tri*-ionic hydrates, which complies well with the calculated *E*_B_ results previously mentioned in Section 3.1.5 and also with an IL-water finding carried out by Danten et al. [91] that the vibrational spectra of [C_n_mim]X (*n* = 1, 2, 4, 8; X = BF_4_^−^, PF_6_^−^) in mixtures with water have only a little correlation with the alkyl chain. The same point also holds for the *tri*- and *tetra*-ionic species in aqueous solutions.

Admittedly, the red-shifted changes in the maxima peak for thymine in T-3H_2_O-[C_n_mim]^+^-Br^−^-1 for the counterpart in an isolated state mainly involve vibrations of the four interaction sites (N1-H, C2=O7, N3-H, and C4=O8), in Figure 5(a4). The four maxima absorption for the slightly more favorable T-3H_2_O-[C_6_mim]^+^-Br^−^-1 are yielding in 3453, 3281, 1749, and 1713 cm^−1^, with the comparison of the T-4H_2_O-Br^−^-1, experiences a respective blue-shift, among which the first two bands are almost determined by the stretching ν(N-H) modes. It implies the formation of new NCIs between thymine and cations at the expense of the old HBs between the base and water molecule, which was pointed out in a reported work [92] on the ILs [C_4_mim]X (X = BF_4_^−^, Cl^−^, Br^−^, I^−^). This interaction which either weakens strengths or the broken bonds for the intrinsic HBs between [C_4_mim]^+^ cation and the anions causes a high-wavenumber shift upon an aqueous dilution. Closer observation of the mainly respective vibrations corresponding to the four maxima, the v7 (100%), v10 (100%), v32 (98%), and v34 (98%), absorption is weakened relative to the T-4H_2_O-Br^−^-1; this indicates the π-π^+^ stacking formation.

Upon the intercalation of the second pair of IL [C_n_mim]Br into the “wobble” site (i.e., C2=O7 and N3-H binding sites), equivalent to the replacement of the two molecules in T-3H_2_O-[C_n_mim]^+^-Br^−^-1. In Figure 5(a5,a6), compared with the T-3H_2_O-[C_6_mim]^+^-Br^−^-1, the amount of redshift from 3453 to 3400 cm^−1^, assigned to the complete ν(N1-H) mode (100%), and the absorption intensity enhanced from 956 to 1245 cm^−1^, result in a stronger intermolecular N1-H···Br^−^ HB arises from the *di*- to *tetra*-ionic microhydrates. However, the peak assigned mainly to the ν(N3-H) mode suffers a blue-shifted wavelength due to the NCI change from the N3-H···Ow to N3-H···Br^−^ HB. The three vibration modes within 1800~1700 cm^−1^ for the T-1H_2_O-2[C_6_mim]^+^-2Br^−^-1 correspond to almost the same wavenumbers as those for T-3H_2_O-[C_6_mim]^+^-Br^−^-1, but the intensities show a visible increment, which is ascribed to the increasing π-π^+^ stacking. Thus, upon the participation of a second ion pair, in an aqueous solution, thymine is still more sensitive to H-bonding environments with Br^−^ anion than stacking and weaker ^+^(C2-H)···O7 ambiances with the [C_n_mim]^+^ cations, which lives up to the conclusion obtained by Leist et al. [93] that intermolecular HBs should always be utilized and the π-π or C-H···π interaction is not dominant NCI once the C-H···O=C hydrogen bonds formed.

### 3.3. UV-Vis Spectra

#### 3.3.1. Isolated Thymine, Cations, and Water Molecule

Figure 5(b1–b3) shows the calculated electronic spectra for monomeric thymine, H_2_O, and the five imidazolium cations with different alkyl chain lengths at the TD-M06-2X/ 6-311++G(2*d*, *p*)/PCM/water level. Obviously, in the ultraviolet region, 220~260 nm, the characteristic peak at 238 nm exclusive to thymine is assigned to the S_0_→S_1_ transition (96%, Figure 5(b2)), which mainly corresponds to the HOMO→LUMO excitation with π/π* type as confirmed by Gustavsson et al. [58,94], and also stated by Shukla et al. [95] that the π/π* state is the lowest singlet excited state in a protic solvent. In the subsequent study, the first excited singlet state, S_1_, will be the electronic state that deserves more attention.

#### 3.3.2. Ionic T-wH_2_O-y[C_n_mim]^+^-xBr^−^-1 Microhydrates

Compared with the monomeric thymine, the absorption spectra lying at 238 nm show a remarkable red shift (bathochromism) with decreasing solvation, which is a consequence of the stepwise formation of more NCIs established between thymine and the ILs [C_n_mim]Br in increasing ionic size, in Table 1, reduce the HOMO-LUMO gap.

In the ordered T-4H_2_O-Br^−^-1, T-3H_2_O-[C_n_mim]^+^-Br^−^-1, T-2H_2_O-[C_n_mim]^+^-2Br^−^-1 (barring T-2H_2_O-[C_2_mim]^+^-2Br^−^-1), and T-1H_2_O-2[C_n_mim]^+^-2Br^−^-1, the S_0_→S_1_ excitation is mainly attributed to the HOMO-2→LUMO (57%), HOMO→LUMO (52~53% for the varying cations), HOMO→LUMO (34~38% for the different cations), and HOMO-3→LUMO (74~78% for the varying cations) excitation, one after another (see Table 1). Nonetheless, among each of the four transitions with n/π* type, the occupied orbital (HOMO/HOMO-1/HOMO-2/HOMO-3) dominates in the lone-pair bromide anion, whereas the LUMO is devoted to the thymine moiety, which has a corresponding transition as illustrated in Appendix A. It implies that the bands at ~239, 243, 242, and 246 nm for these decreasing hydrated species involve charge transfer from the Br^−^ anion to the delocalized thymine. This occurrence is independent of the alkyl chain lengths of the imidazolium cations, as discussed above and shown in Figure 5(b4–b6).

It is interesting to note that in addition to the bathochromism, with the hampered hydration, the hypochromic effect has existed in the T-*w*H_2_O-*y*[C_n_mim]^+^-*x*Br^−^-1 and is more remarkable when the ion clusters grow in size following orderly [C_n_mim]^+^-Br^−^, [C_n_mim]^+^-2Br^−^, and 2[C_n_mim]^+^-2Br^−^. It is associated with the π-π^+^ stacking interaction between the pyrimidine ring and the cationic imidazolium ring, as bourne out by Pérez-Flores et al. [96]. They are dedicated to observing the acridine-spermine conjugate’s spectral changes in the presence of different oligonucleotide concentrations. On a closer comparison of the tri-hydrated T-3H_2_O-[C_n_mim]^+^-Br^−^-1 and mono-hydrated T-1H_2_O-2[C_n_mim]^+^-2Br^−^-1 species for the given n, the S_0_ → S_1_ transition of hydrates having a two ion pair is redshifted by as much as ~3 nm from that of T-3H_2_O-[C_n_mim]^+^-Br^−^-1 with decreasing oscillator strengths (~0.02). Thus, the participation of the second ion pair permits a stronger H-bonding and stacking environment conferred by thymine and the imidazolium of IL.

### 3.4. QTAIM Analysis

The electronic structure analysis employing QTAIM and NBO methodologies were performed to explore interactions between thymine and the 1-alkyl-3-methylimidazolium bromide ionic liquids [C_n_mim]Br (*n* = 2, 4, 6, 8, 10) in essence. As a local descriptor, the QTAIM, more precise criteria compared with the geometric respect, is employed to describe the H-bonding and van der Waals NCIs [97]. Given that an insignificant discrepancy could always be observed with varying cationic alkyl chain lengths, no matter whether from a geometrical or spectral perspective, only the topography pictures of the T-4H_2_O-Br^−^-1, T-3H_2_O-[C_6_mim]^+^-Br^−^-1, T-2H_2_O-[C_6_mim]^+^-2Br^−^-1, and T-1H_2_O-2[C_6_mim]^+^-2Br^−^-1 hydrates are shown in Figure 6. Generally, the description of the nature and strength of NCIs is dependent on the topological properties of (3, −1) bond critical point (BCP) [98,99] and (3, +3) cage critical point (CCP) [100]. Table 2 lists the following descriptors at BCPs and CCPs that exist between the respective T-Br^−^, T-[C_6_mim]^+^, and [C_6_mim]^+^-Br^−^ fragments: electron density *ρ*(*r*) and its gradient Laplacian ∇^2^*ρ*(*r*), the kinetic *G*(*r*), potential *V*(*r*), and total *H*(*r*) energy densities. ∇^2^*ρ*(*r*) > 0 is characterized by a closed-shell interaction (including H-bonding and π-stacking), in which the *ρ*(*r*) value is typically small or, more specifically, ~0.01 a.u. or less for H-bonding interaction and ~0.001 a.u. for van der Waals contact [101].

Additionally, the corresponding HB energies (*E*_HB_, kJ/mol) calculated as Equation (4) are also represented.
(4)$$E_{\mathrm{HB}}=\frac{1}{2}V\left(r\right)$$

For the four hydrates, *ρ*(*r*) values show the main H-bond CP, N-H···Br^−^ between the T-Br^−^ moiety, which is characteristic of electron density and its Laplacian ranging within 0.018~0.022 a.u and 0.050~0.055 a.u, respectively, and is closer to the standard range (*ρ*(*r*) = 0.002~0.040 and ∇^2^*ρ*(*r*) = 0.024~0.139 a.u) proposed by Koch and Popelier [102,103]. The N1-H···Br^−^ topological properties between T-3H_2_O-[C_6_mim]^+^-Br^−^-1 and T-1H_2_O-2[C_6_mim]^+^-2Br^−^-1 show a greater *ρ*(*r*) value for the latter. In the case of interactions between cation-anion, some of the corresponding *ρ*(*r*) values are small enough to be distinguishd only by the rather rough geometrics. Table 2 shows that the amount of CPs increases with growing ion clusters, which agrees with the higher stabilities of the *tetra*-ionic T-1H_2_O-2[C_n_mim]^+^-2Br^−^-1 hydrates than the *tri*-, *di*-, and *mono*-ionic species. It seems plausible that the *ρ*_BCP_ values have negative correlations with the corresponding H-bond distances while positive ones have the π···π^+^ distance of centroids.

Figure 6 shows four BCPs and seven CCPs in the T-3H_2_O-[C_6_mim]^+^-Br^−^-1 hydrate and seven BCPs and five CCPs in the T-1H_2_O-2[C_6_mim]^+^-2Br^−^-1 used to describe π-π^+^ interactions. The sum of *ρ*_BCP_ and *ρ*_CCP_ (Ʃ*ρ*_BCP_ and Ʃ*ρ*_CCP_) and the sum of corresponding Laplacian (Ʃ∇^2^*ρ*_BCP_ and Ʃ∇^2^*ρ*_CCP_) are calculated to compare the π-π^+^ interaction between T-3H_2_O-[C_6_mim]^+^-Br^−^-1 and T-1H_2_O-2[C_6_mim]^+^-2Br^−^-1. It transpires that the Ʃ*ρ*_BCP_ and Ʃ∇^2^*ρ*_BCP_ between two aromatic π rings in *tetra*-ionic hydrate are greater than those in *di*-ionic counterpart, although the Ʃ*ρ*_CCP_ and Ʃ∇^2^*ρ*_CCP_ are almost equal for the two systems (0.048 and 0.168 a.u. for T-1H_2_O-2[C_6_mim]^+^-2Br^−^-1, 0.029 and 0.015 for T-3H_2_O-[C_6_mim]^+^-Br^−^-1). It indicates that the π-π^+^ interaction between thymine and the imidazolium cation is much stronger upon adding two equivalents of [C_6_mim]Br. Figure 7a–c) show an excellent linear correlation, indicated by the regression coefficients in parentheses, between *E*_HB_ and the *ρ*(*r*) values at BCPs for the T-3H_2_O-[C_6_mim]^+^-Br^−^-1 (0.933), T-2H_2_O-[C_6_mim]^+^-2Br^−^-1 (0.960), and T-1H_2_O-2[C_6_mim]^+^-2Br^−^-1 (0.966) hydrates, orderly, which signifies cooperative effect between intermolecular T-Br^−^, T-[C_6_mim]^+^, and [C_6_mim]^+^-Br^−^ fragments incrementally present along with the increasing ion clusters, and supports a linear dependence between *E*_B_ values and increasing ionic number as discussed before.

### 3.5. NBO Population Analysis

The natural population analysis (NPA) using NBO method was also performed to focus on the atomic charges and orbital interaction energy for the studied T-*w*H_2_O-*y*[C_n_mim]^+^-*x*Br^−^-1 hydrates. The stabilization of orbital interaction (*E*^2^) is calculated charge transfer energies between the Lewis type NBO’s and non-Lewis type NBO’s, followed by Equation (5).
(5)$$E^{2}=\frac{q_{i}\cdot F_{ij}^{2}}{\varepsilon_{i}-\varepsilon_{j}}$$
where *q_i_* is the donor orbital occupancy, $F_{ij}^{2}$ is the off-diagonal NBO Fock matrix element, and *ε_i_* and *ε_j_* are the diagonal elements (orbital energies) of the donor (*i*) and acceptor (*j*). Generally, the higher the *E*^2^ values, the greater the stabilities of the targeted complexes [104]. Especially, the sum of donor-acceptor interaction energy (∑*E*^2^ = *E*^2^_π→__π+_ + *E*^2^_π←__π+_) between thymine and the imidazolium cations [C_n_mim]^+^ (*n* = 2, 4, 6, 8, 10) is used to describe as the π-π^+^ interaction. According to the NPA charges in Appendix A, the neutral thymine is more inclined to act as an electron acceptor, and the interactions with more than 2 kJ/mol energy [105] between the base and [C_n_mim]Br in aqueous solutions are listed in Table 3. The highest stabilization energy corresponds to the interaction between the lone pair electron of Br^−^ (lpBr^−^) and 2-center antibond (BD*) N-H orbitals, which orientates unequivocally to the lpBr^−^→σ*N1-H for the T-4H_2_O-Br^−^-1 and T-3H_2_O-[C_n_mim]^+^-Br^−^-1 while the lpBr^−^→σ*N3-H for the T-2H_2_O-[C_n_mim]^+^-2Br^−^-1 and T-1H_2_O-2[C_n_mim]^+^-2Br^−^-1. Of the greatest contributor in per ionic thymine hydrate, the specific value shows 74.99, 40.63~45.81 (for varying n), 68.13~69.01, and 75.28~76.33 kJ/mol *E*^2^ energy in the respective T-4H_2_O-Br^−^-1, T-3H_2_O-[C_n_mim]^+^-Br^−^-1, T-2H_2_O-[C_n_mim]^+^-2Br^−^-1, and T-1H_2_O-2[C_n_mim]^+^-2Br^−^-1 hydrates. Comparison of the *di*-ionic hydrates, the amount of NPA charge transfer doubles in the *tetra*-ionic species, where the π-π^+^ stacking characterized by the ∑*E*^2^ values shows 37.24~39.12 kJ/mol for the varying cations and is more than that in T-3H_2_O-[C_n_mim]^+^-Br^−^-1 showing 20.82~21.69 kJ/mol. These calculated results coincide with geometrics, energetics, spectroscopy, and AIM analysis.

## 4. Conclusions

In this work, the density functional theory method M06-2X/6-311++G(2*d*, *p*) was used to calculate the three typical interaction modes of cation and two anionic interaction modes between1-alkyl-3-methylimidazolium bromide ([C_n_mim]Br, *n* = 2, 4, 6, 8, 10) ionic liquids to the thymine (T) in aqueous environment. At the same time, the effect of the first solvation shell was investigated. The geometric structure, energetics, spectral characteristics and electronic structure were compared and analyzed, and the hydrogen bond interaction and π-π interaction in the complxes were explored in detail.

***PCM Model***. The relative and binding energies (*E*_R_ and *E*_B_) show the same trend for the favorable arrangements of thymine with the cations (T-[C_n_mim]^+^): stacking > perpendicular > coplanar and with the Br^−^ anion (T-Br^−^): front > top. The interaction of thymine with [C_n_mim]^+^ cations presents a weaker base-ion strength than that with Br^−^ anion, irrespective of the growing ion clusters increasing overall stabilities or the progressive solvation.

***Solvation Model***. Upon the addition of water-explicit molecule(s) gradually, the relative stabilities continue following: T-5H_2_O-1 < T-4H_2_O-Br^−^-1 < T-3H_2_O-[C_n_mim]^+^-Br^−^-1 < T-2H_2_O-[C_n_mim]^+^-2Br^−^-1 < T-1H_2_O-2[C_n_mim]^+^-2Br^−^-1. A good linear-like relationship between *E*_B_ values and the increasing number (*x* + *y*) of ions indicates the cooperative effect between H-bonding and π-π^+^ stacking. Additionally, with the increasing lengths of the cationic alkyl chain, regardless of the *di*- or *tetra*-ionic hydrates, n equals 6, which is more favored energetically, albeit slightly.

To the isolated thymine (T), the IR and UV-Vis spectra for the microhydrated *mono*-, *di*-, *tri*-, and *tetra*-ionic thymine suggest that the formation of N-H···Br^−^ HB between thymine and the anion results in more significant redshifts and incrementing strengths (bathochromism). In contrast, π-π^+^ stacking interaction between thymine and the imidazolium cations leads to less critical shifts with varying alkyl chain length and decrementing strengths (hypochromism).

The QTAIM and NBO calculations suggest that the cooperativity effect exists between T-Br^−^, T-[C_n_mim]^+^, and [C_n_mim]^+^-Br^−^ fragments and fits well with the high linear correlation between *E*_B_ values and increasing ionic number; the *E*^2^ value of lpBr^−^→σ*N-H is the maximum contribution to the charge transfer energies and is well-beyond the ∑*E*^2^ values for the sum of *E*^2^_π→__π+_ and *E*^2^_π←__π+_.

## Figures and Tables

**Figure 1 molecules-27-06242-f001:**
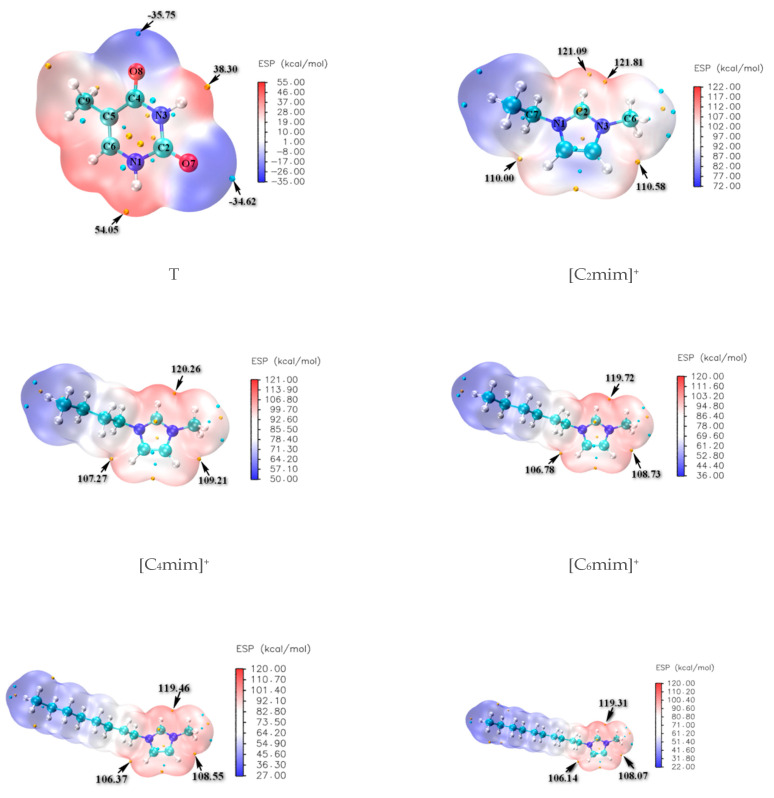
Molecular electrostatic potential (MEP) maps of the thymine (T) and imidazolium cations -[C_n_mim]^+^ (*n* = 2, 4, 6, 8, 10).

**Figure 2 molecules-27-06242-f002:**
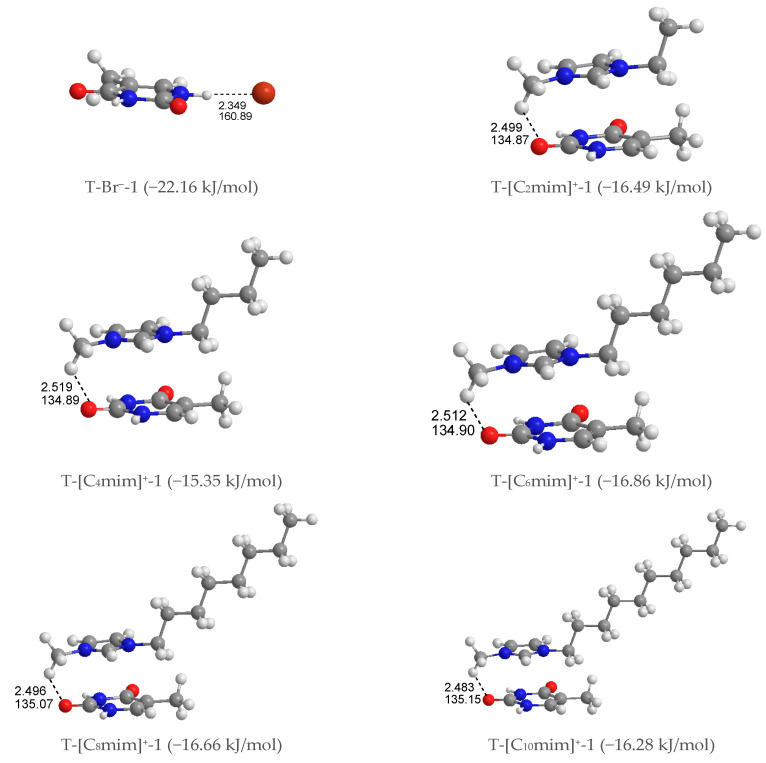
The most stable structures for the mono-ionic thymine, T-Br^−^-1 and T-[C_n_mim]^+^-1 (*n* = 2, 4, 6, 8, 10). The bond length in Å, H-bond angle in °, binding energy listed in parentheses.

**Figure 3 molecules-27-06242-f003:**
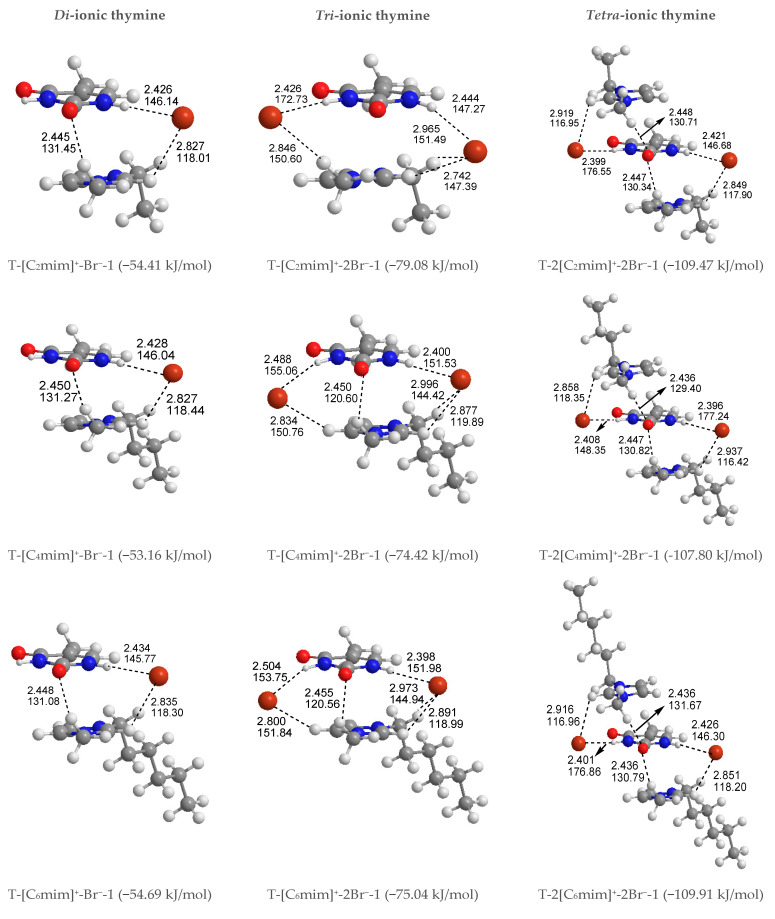
The most stable structures for the free-, *mono*-, *di*-, *tri*-, and *tetra*-ionic microhydrates, T-5H_2_O-1, T-4H_2_O-Br^−^-1, T-3H_2_O-[C_n_mim]^+^-Br^−^-1, T-2H_2_O-[C_n_mim]^+^-2Br^−^-1, and T-1H_2_O-2[C_n_mim]^+^-2Br^−^-1 (*n* = 2, 4, 6, 8, 10). The bond length in Å, H-bond angle in °, the binding energy listed in parentheses. Ionic T-wH_2_O-y[C_n_mim]^+^-xBr^−^-k microhydrates.

**Figure 4 molecules-27-06242-f004:**
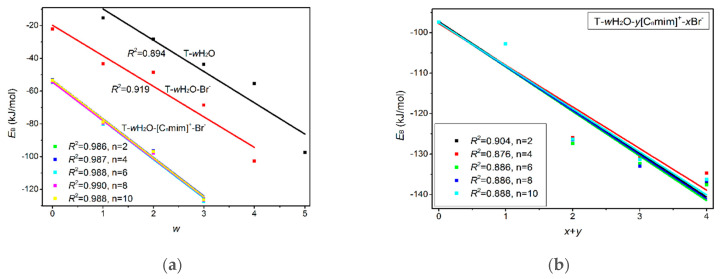
(**a**) Correlations between the *E*_B_ and number of the explicit water molecule (*w*) for the microhydrates, T-*w_1_*H_2_O-Br^−^-1, T-*w_2_*H_2_O-[C_n_mim]^+^-Br^−^-1, T-*w_3_*H_2_O-[C_n_mim]^+^-2Br^−^-1, and T-*w_4_*H_2_O-2[C_n_mim]^+^-2Br^−^-1 (*n* = 2, 4, 6, 8, 10). (**b**) Correlations between the *E*_B_ and number of the ion (*x*+*y*) for the microhydrates, T-4H_2_O-Br^−^-1, T-3H_2_O-[C_n_mim]^+^-Br^−^-1, T-2H_2_O-[C_n_mim]^+^-2Br^−^-1, and T-1H_2_O-2[C_n_mim]^+^-2Br^−^-1 (*n* = 2, 4, 6, 8, 10).

**Figure 5 molecules-27-06242-f005:**
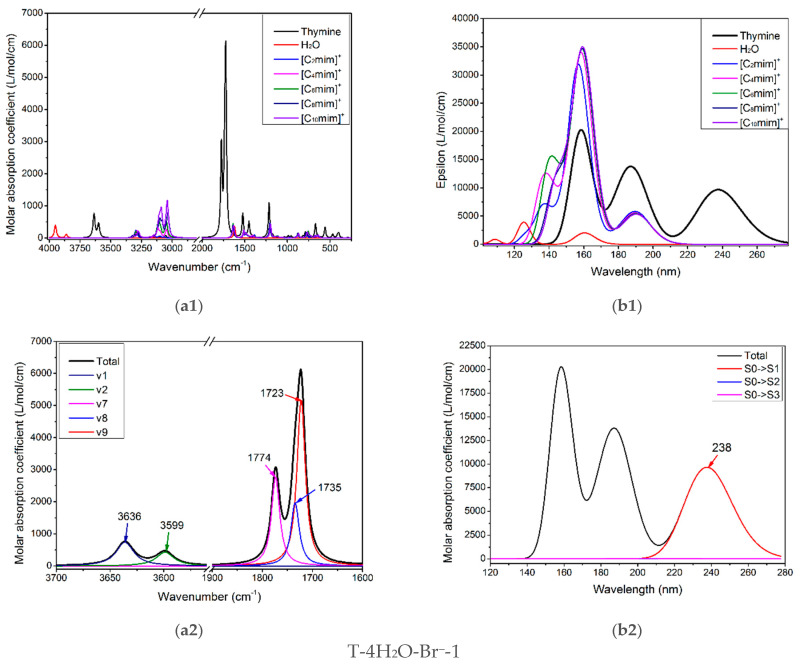
(**a1**–**a6**) The IR vibrational spectra and (**b1**–**b6**) UV-Vis absorption spectra of the thymine monomer and the corresponding microhydrates, T-4H_2_O-Br^−^-1, T-3H_2_O-[C_n_mim]^+^-Br^−^-1, T-2H_2_O-[C_n_mim]^+^-2Br^−^-1, and T-1H_2_O-2[C_n_mim]^+^-2Br^−^-1 (*n* = 2, 4, 6, 8, 10). The modes v1, v2, v7, v8, and v9 dominating in absorption peaks located at 3636, 3600, 1774, and 1723 cm^−1^ are assigned to νN1-H, νN3-H, υC2=O7 + ρN1-H, υC4=O8 + ρN3-H, and υC5=C6 + ρN3-H + ρC6-H for the isolated thymine, in order.

**Figure 6 molecules-27-06242-f006:**
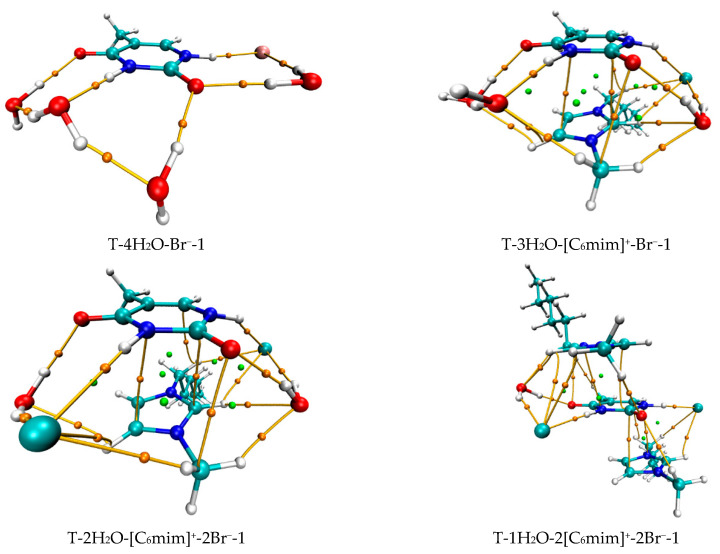
Topological properties calculated at intermolecular BCPs, and CCPs between thymine-ILs in the microhydrated T-4H_2_O-Br^−^-1, T-3H_2_O-[C6mim]^+^-Br^−^-1, T-2H_2_O-[C_6_mim]^+^-2Br^−^-1, and T-1H_2_O-2[C_6_mim]^+^-2Br^−^-1 using AIM theory.

**Figure 7 molecules-27-06242-f007:**
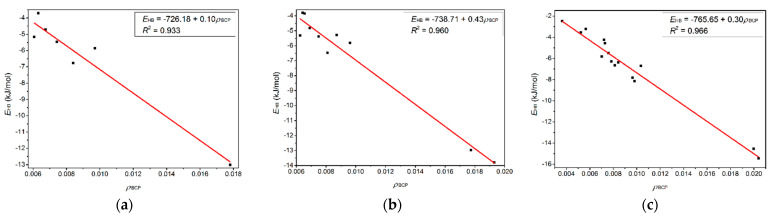
Correlations variation of the hydrogen bond energy (*E*_HB_, kJ/mol) with the electron density at BCPs (*ρ*_BCP_, a.u.) for the microhydrates, (**a**) T-3H_2_O-[C_6_mim]^+^-Br^−^-1, (**b**) T-2H_2_O-[C_6_mim]^+^-2Br^−^-1, and (**c**) T-1H_2_O-2[C_6_mim]^+^-2Br^−^-1.

**Table 1 molecules-27-06242-t001:** TD-DFT calculated absorption maxima wavenumber (l_max_, nm), oscillator strength (*f*), transition electronic orbitals, transition assignment, and the corresponding main contributions (%) to the absorption maxima located within 220–280 nm for the microhydrates, T-4H_2_O-Br^−^-1, T-3H_2_O-[C_n_mim]^+^-Br^−^-1, T-2H_2_O-[C_n_mim]^+^-2Br^−^-1, and T-1H_2_O-2[C_n_mim]^+^-2Br^−^-1 (*n* = 2, 4, 6, 8, 10).

Complexes	λ_max_	*f*	Transition	Assignment	Contribution
T	237.41	0.2390	H → L	π/π*	96
T-4H_2_O-Br^−^-1	238.94	0.2931	H-2 → L	n/π*	57
T-3H_2_O-[C_2_mim]^+^-Br^−^-1	243.09	0.2030	H → L	n/π*	53
T-3H_2_O-[C_4_mim]^+^-Br^−^-1	243.02	0.1991	H → L	n/π*	53
T-3H_2_O-[C_6_mim]^+^-Br^−^-1	242.93	0.1984	H → L	n/π*	52
T-3H_2_O-[C_8_mim]^+^-Br^−^-1	243.00	0.2005	H → L	n/π*	53
T-3H_2_O-[C_10_mim]^+^-Br^−^-1	243.00	0.2002	H → L	n/π*	53
T-2H_2_O-[C_2_mim]^+^-2Br^−^-1	242.76	0.1897	H-1 → L	n/π*	35
T-2H_2_O-[C_4_mim]^+^-2Br^−^-1	242.54	0.1874	H → L	n/π*	38
T-2H_2_O-[C_6_mim]^+^-2Br^−^-1	242.54	0.1872	H → L	n/π*	34
T-2H_2_O-[C_8_mim]^+^-2Br^−^-1	242.56	0.1885	H → L	n/π*	34
T-2H_2_O-[C_10_mim]^+^-2Br^−^-1	242.57	0.1867	H → L	n/π*	35
T-1H_2_O-2[C_2_mim]^+^-2Br^−^-1	246.64	0.1816	H-3 → L	n/π*	75
T-1H_2_O-2[C_4_mim]^+^-2Br^−^-1	246.47	0.1819	H-3 → L	n/π*	77
T-1H_2_O-2[C_6_mim]^+^-2Br^−^-1	246.61	0.1816	H-3 → L	n/π*	74
T-1H_2_O-2[C_8_mim]^+^-2Br^−^-1	246.52	0.1832	H-3 → L	n/π*	74
T-1H_2_O-2[C_10_mim]^+^-2Br^−^-1	246.63	0.1819	H-3 → L	n/π*	78

Note: H-HOMO, L-LUMO.

**Table 2 molecules-27-06242-t002:** Topological parameters (a.u.) obtained from the AIM analysis for the T-4H_2_O-Br^−^-1, T-3H_2_O-[C_6_mim]^+^-Br^−^-1, T-2H_2_O-[C_6_mim]^+^-2Br^−^-1, and T-1H_2_O-2[C_6_mim]^+^-2Br^−^-1 microhydrates at the BCPs and CCPs. ƩBCP and ƩCCP characterize the π-π^+^ interactions.

Interaction	CPs	*R*_X···Y_(Å)	*ρ*(*r*)	∇^2^*ρ*(*r*)	*G*(*r*)	*V*(*r*)	*H*(*r*)	*E*_HB_ (kJ/mol)
		T-4H_2_O-Br^−^-1						
T-Br^−^	N1-H···Br^−^	2.352	0.022	0.057	0.014	−0.013	0.001	−17.25
		T-3H_2_O-[C_6_mim]^+^-Br^−^-1						
T-Br^−^	N1-H···Br^−^	2.482	0.018	0.050	0.011	−0.010	0.001	−13.01
T-[C_6_mim]^+^	ƩBCP ^a^	3.327	0.029	0.105	0.021	−0.017	0.005	-
	ƩCCP ^b^	-	0.026	0.107	0.022	−0.017	0.005	-
[C_6_mim]^+^-Br^−^	^+^C(2)-H···Br^−^	3.101	0.006	0.016	0.003	−0.003	0.001	−3.69
	^+^C(7)-H···Br^−^	2.843	0.010	0.026	0.005	−0.004	0.001	−5.86
		T-2H_2_O-[C_6_mim]^+^-2Br^−^-1						
T-Br^−^	N1-H···Br^−^	2.486	0.018	0.050	0.011	−0.010	0.001	−12.96
	N3-H···Br^−^	2.414	0.019	0.052	0.012	−0.011	0.001	−13.79
T-[C_6_mim]^+^	ƩBCP ^a^	3.327	0.029	0.103	0.021	−0.017	0.005	-
	ƩCCP ^b^	-	0.026	0.101	0.021	−0.016	0.004	-
[C_6_mim]^+^-Br^−^	^+^C(2)-H···Br^−^	3.079	0.010	0.025	0.005	−0.004	0.001	−5.80
	^+^C(4)-H···Br^−^	3.176	0.006	0.018	0.004	−0.003	0.001	−3.81
	^+^C(6)-H···Br^−^	2.903	0.009	0.023	0.005	−0.004	0.001	−5.28
	^+^C(7)-H···Br^−^	2.851	0.007	0.017	0.004	−0.003	0.001	−3.84
		T-1H_2_O-2[C_6_mim]^+^-2Br^−^-1						
T-Br^−^	N1-H···Br^−^	2.408	0.020	0.055	0.013	−0.012	0.001	−15.43
	N3-H···Br^−^	2.398	0.020	0.053	0.012	−0.011	0.001	−14.53
T-[C_6_mim]^+^	ƩBCP ^a^	3.386	0.048	0.168	0.035	−0.028	0.007	-
	ƩCCP ^b^	-	0.022	0.087	0.018	−0.015	0.004	-
	^+^C(6)-H···O7	2.458	0.010	0.032	0.007	−0.006	0.001	−7.83
	^+^C(6)-H···O7	2.444	0.010	0.033	0.007	−0.006	0.001	−8.17
[C_6_mim]^+^-Br^−^	^+^C(2)-H···Br^−^	3.298	0.006	0.016	0.003	−0.002	0.001	−3.23
	^+^C(2)-H···Br^−^	2.856	0.010	0.030	0.006	−0.005	0.001	−6.71
	^+^C(6)-H···Br^−^	3.091	0.007	0.021	0.004	−0.003	0.001	−4.54
	^+^C(7)-H···Br^−^	3.045	0.007	0.018	0.004	−0.003	0.001	−4.23

^a^ the sum of *ρ*_BCP_ and corresponding Laplacian ∇^2^*ρ*_BCP_ (Ʃ*ρ*_BCP_ and Ʃ∇^2^*ρ*_BCP_); ^b^ the sum of *ρ*_CCP_ and corresponding Laplacian ∇^2^*ρ*_CCP_ (Ʃ*ρ*_CCP_ and Ʃ∇^2^*ρ*_CCP_).

**Table 3 molecules-27-06242-t003:** The donor-acceptor interaction energies (*E*^2^, kJ/mol) obtained from the NBO analysis on the wave functions for the microhydrates, T-4H_2_O-Br^−^-1, T-3H_2_O-[C_n_mim]^+^-Br^−^-1, T-2H_2_O-[C_n_mim]^+^-2Br^−^-1, and T-1H_2_O-2[C_n_mim]^+^-2Br^−^-1 (*n* = 2, 4, 6, 8, 10).

T-4H_2_O-Br^−^-1	T-Br^−^					
Interaction	lpBr → σ* N1-H					
	74.99					
Interaction	T-Br^−^	T-[C_n_mim]^+^		[C_n_mim]^+^-Br^−^		
	lpBr^−^ → σ*N1-H	π→π^+^	π←π^+^	lpBr^−^→ σ*^+^C(7)-H	lpBr^−^→ σ*^+^C(2)-H	
T-3H_2_O-[C_2_mim]^+^-Br^−^-1	45.81	12.83	8.86	12.41	6.35	
T-3H_2_O-[C_4_mim]^+^-Br^−^-1	41.30	12.29	9.11	14.13	7.73	
T-3H_2_O-[C_6_mim]^+^-Br^−^-1	40.63	11.45	9.41	14.21	7.69	
T-3H_2_O-[C_8_mim]^+^-Br^−^-1	43.43	11.62	9.20	14.21	7.15	
T-3H_2_O-[C_10_mim]^+^-Br^−^-1	43.22	11.66	9.07	14.21	6.94	
Interaction	T-Br^−^		T-[C_n_mim]^+^			
	lpBr^−^ → σ*N3-H	lpBr→ σ*N1-H	π→π^+^	π←π^+^		
T-2H_2_O-[C_2_mim]^+^-2Br^−^-1	68.47	43.14	12.75	7.77		
T-2H_2_O-[C_4_mim]^+^-2Br^−^-1	68.84	39.29	11.83	7.86		
T-2H_2_O-[C_6_mim]^+^-2Br^−^-1	68.13	40.04	11.70	8.11		
T-2H_2_O-[C_8_mim]^+^-2Br^−^-1	69.01	41.97	11.70	8.11		
T-2H_2_O-[C_10_mim]^+^-2Br^−^-1	68.38	40.13	11.87	8.07		
Interaction	[C_n_mim]^+^-Br^−^					
	lpBr^−^→ σ*^+^C(7)-H	lpBr^−^ → σ*^+^C(6)-H	lpBr^−^ → σ*^+^C(2)-H		lpBr^−^→ σ*^+^C(4)-H	
T-2H_2_O-[C_2_mim]^+^-2Br^−^-1	12.16	8.03	6.98	2.09		
T-2H_2_O-[C_4_mim]^+^-2Br^−^-1	14.13	8.53	7.69	-		
T-2H_2_O-[C_6_mim]^+^-2Br^−^-1	13.75	8.19	8.15	2.05		
T-2H_2_O-[C_8_mim]^+^-2Br^−^-1	13.88	8.49	7.69	-		
T-2H_2_O-[C_10_mim]^+^-2Br^−^-1	13.63	7.98	7.98	2.05		
Interaction	T-Br^−^		T-[C_n_mim]^+^			
	lpBr^−^→ σ*N3-H	lpBr^−^→ σ*N1-H	lpO7→ σ*^+^C(6)-H		π←π^+^	π→π^+^
T-1H_2_O-2[C_2_mim]^+^-2Br^−^-1	76.33	60.07	2.38		10.95	27.80
T-1H_2_O-2[C_4_mim]^+^-2Br^−^-1	75.62	57.10	2.26		11.12	27.63
T-1H_2_O-2[C_6_mim]^+^-2Br^−^-1	76.03	57.18	2.68		10.99	26.25
T-1H_2_O-2[C_8_mim]^+^-2Br^−^-1	75.28	58.44	2.34		11.20	26.92
T-1H_2_O-2[C_10_mim]^+^-2Br^−^-1	75.99	57.39	2.34		11.20	27.92
Interaction	[C_n_mim]^+^-Br^−^					
	lpBr^−^→ σ*^+^C(2)-H	lpBr^−^→ σ*^+^C(7)-H	lpBr^−^→π* ^+^C(2)-N3		lpBr^−^→σ*^+^C(6)-H	
T-1H_2_O-2[C_2_mim]^+^-2Br^−^-1	7.94	5.48	4.97		-	
T-1H_2_O-2[C_4_mim]^+^-2Br^−^-1	8.23	5.52	4.81		4.60	
T-1H_2_O-2[C_6_mim]^+^-2Br^−^-1	8.36	6.31	4.72		2.34	
T-1H_2_O-2[C_8_mim]^+^-2Br^−^-1	8.19	5.89	4.89		4.22	
T-1H_2_O-2[C_10_mim]^+^-2Br^−^-1	8.19	5.14	4.93		2.26	

## Data Availability

All data for this work are provided in the manuscript.

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
