# Peer review of "Density Functional Method Study on the Cooperativity of Intermolecular H-bonding and π-π+ Stacking Interactions in Thymine-[Cnmim]Br (n = 2, 4, 6, 8, 10) Microhydrates"

_molecules, 2022, doi:10.3390/molecules27196242_

Round 1
Reviewer 1 Report
Brief Summary: After a careful study of the manuscript, I would like to recommend the manuscript may be publishable in the journal “Molecules”. This work reports a well written, observations are completely based on theoretical analysis. The observations are innovative. The computational supports are adequate. I recommend the article to be accepted in its present form.
Recommendation: Accept.
Author Response
Dear Reviewer,
Thank you for taking time out of your busy schedule to review the manuscript (molecules-1891960).
We are delighted that you have recognized the work of theoretical calculation in studying the hydration system of ionic liquids and nucleobase thymine.
We will work harder to do an excellent job in the follow-up system research and look forward to your review and guidance again.
Thanks again
Sincerely
Chaojie Wang
Reviewer 2 Report
In the manuscript: "Density Functional Method Study on the Cooperativity of Intermolecular H-bonding and π-π+ Stacking Interactions in Thymine-[Cnmim]Br (n = 2, 4, 6, 8, 10) Microhydrates" authors explored the binding of different 1-alkyl-3-methylimidazolium bromide ionic liquids to the thymine molecule in a water environment and a microhydrated surrounding. The authors determined the most favorable mutual arrangement of thymine and studied cations/anions and microhydrates. Authors also reported that a good relationship between binding energies and the increasing number of ions was found. In my opinion, the results of the study are interesting for the field of noncovalent interactions and the methodology used in this study is satisfactory. However, there are certain minor suggestions from my side to improve the manuscript before publication:
1. I suggest that authors analyze geometries of noncovalent interactions in the crystal structure of thymine molecule and compare them to their theoretical results. I do not insist on this (it is only a suggestion) but I believe that the results of the suggested analysis would additionally strengthen the conclusions from this study.
2. Authors missed citing an important study related to the non-covalent interactions of thymine molecule: CrystEngComm, 2014,16, 10089-10096. In the suggested study electrostatic potential map of thymine was calculated and used for the explanation of the calculated energies of weak noncovalent interactions involving thymine molecule and water. I believe that these results are closely related to the results presented in the submitted manuscript and that it is necessary to include them in the introduction.
Author Response
Dear Reviewer,
Thank you for taking time out of your busy schedule to review the manuscript (molecules-1891960).
Thank you very much for affirming theoretical calculation research on ionic liquids and nucleobase thymine hydration systems. Your comments and suggestions are of great value to further improve and enhance this work and have been correspondingly revised in the new version.
- For the overall evaluation of the manuscript, "Are the results clearly presented" needs to be improved. We adjusted the abstract and conclusion to reflect the logical order of the research system design, supplemented the schematic diagram of interactionmodes, and explained the definitions of three modes(Stacking, Perpendicular, and Co-planar). In the conclusion section, a paragraph is added to describe the research work. At the same time, the fuzzy expression in the conclusion part is corrected, such as x + y, so that the reader can better understand this work.
- Thank you for your two specific suggestions. First, the crystal structure thymine data is compared with the theoretical calculation. The data comparison in Table S1 is only to show that the structural parameters in vacuum of thyminecalculated by our applied method M06-2X/6-311++G(2d, p) are close to the measured values, which is also one of the ways used in a conventional theoretical study to prove the reliability of the method selected. Of course, the validity of the selection method should be illustrated by other experimental results, such as the coincidence of spectral data. After all, only limited structural data are used. At the same time, ionic liquid interacts with nucleobases in an aqueous environment, and the related structures are rarely reported.
The second piece of advice is precious. It has been added in the introduction. This reference paper is typical for the analysis and discussion on NCI, which is worthy of learning. Thanks to the reviewer for other advice, reminding us to more extensive review of the literature on the subject and a more thorough analysis of the research.
Thanks again.
Best Wishes!
Chaojie Wang
Reviewer 3 Report
This paper discusses hydrogen bonding, π-π+ stacking interactions with thymine/alkyl-substituted imidazolium ionic liquids/water molecules using DFT calculations by Wang et al.
As mentioned in the introduction, with the increase in the use of ionic liquids, the interpretation of their affinity for living organisms has become an important issue. I believe that this paper should be published in Molecules
Major issues:
The content of Section 3.1.6 needs to be reviewed as a whole.
The first mentions T-wH2Ohydrates. Here, the interaction between thymine and water molecules 1 to 5 was calculated, but the SI data was only for w = 1 to 4. On the other hand, Figure 4(a) shows data for w = 1 to 5. It is necessary to add w=5 data in SI.
The horizontal axis of Figure 4(a) and (b) is the number of water molecules. Using the dimensionless horizontal axis, you are discussing the additivity by the R2 values. A certain amount of energy is added to EB on the vertical axis as the number of water molecules increases. Therefore, correlation in Figures 4(a) and (b) is natural. Furthermore, I cannot understand the meaning of evaluating these R2 values. The slope and intercept values in Figure 4(a) and (b) are also meaningless. At least the horizontal axis should be a value with dimensions, and their correlations should be sought and discussed.
Isn't the unit of EHB in Table 2 kJ/mol? The Table caption defines it as a.u.
Minor issues:
Line 71: base -> bases?
Line 159: predicated -> predicted?
Line 470: out Pérez-Flores -> out by Pérez-Flores
Line 490: showed -> shown
Lines 619 and 622: Chemosphere. -> Chemosphere
Line 768: add doi:10.3891/acta.chem.scand.53-0057.
Line 789: Rejnek, J.; Hanus, M.; Kabelác, M.; Ryjáček, F.; Shimizu, S.; Hobza, P. -> Check authors order!
Line 795: F. H. C. -> F.H.C.
Author Response
Dear Reviewer,
First, thank you for carefully reviewing the manuscript (molecules-1891960).
In response to your suggestions, we have reorganized the abstract, conclusions, and section 3.1.6, and carefully read and modified it. The whole text clarifies the design ideas and the order of content presentation of this work, and clearly reflects that the hydrogen bond and π-π in the thymine and ionic liquid interaction systems are strongest combined with each other.
Given the main problems, the following modifications are made.
- Insection 1.6, combined with the supporting ESI data, supplementary analysis of the gradual evolution of the computing system, energy, and structural changes in the text, so that readers can better understand the results of our research and the laws found.
- In ESI Figure S6, we have supplemented the structure and binding energy data of the T-5H2O-1 system to make the SI data more complete, thank you for the suggestion.
- Regarding a and b in Figure 4, it is only for the sake of linear fitting the relationship between the binding energy value of each system and the change of water molecules in the system. Then examine the evolution of the carbon number of the alkylchain in the ionic liquid cation at the same time, the overall linear relationship is better, and the binding energy is less affected by the alkyl length. Figure 4 is a fit within defined water numbers, carbon numbers, and ion numbers and does not have the value of the unlimited extension. Considering the dimension, since the abscissa is a dimension of 1, then the slope and intercept are to have a value in unit kJ/mol, to consider whether there is a linear relationship and how the linearity is described by the correlation coefficient R2, without giving the detailed equation itself, the intention is to find that the combination can have a specific range of linear relationship with the number of water molecules and the number of ions.
- In Table 2, it is inappropriate to indicate a.u. directly. To the RX...Y unit is Angstrom, and the EHB is kJ/mol, which has been added in Table 2. Thanks for correcting the error.
- Some other minor bugs. Very sorry we were not careful enough. We have corrected these minor errors one by one. Thank you for pointing this out.
Thanks again!
Sincerely
Chaojie Wang